# SEARCHING FOR DIFFICULT-TO-TRANSLATE TEST EXAMPLES AT SCALE

## ABSTRACT

NLP models require test data that are sufficiently challenging. The difficulty of an example is linked to the topic it originates from ("seed topic"). The relationship between the topic and the difficulty of its instances is stochastic in nature: an example about a difficult topic can happen to be easy, and vice versa. At the scale of the Internet, there are tens of thousands of potential topics, and finding the most difficult one by drawing and evaluating a large number of examples across all topics is computationally infeasible. We formalize this task and treat it as a multi-armed bandit problem. In this framework, each topic is an "arm," and pulling an arm (at a cost) involves drawing a single example, evaluating it, and measuring its difficulty. The goal is to efficiently identify the most difficult topics within a fixed computational budget. We illustrate the bandit problem setup of finding difficult examples for the task of machine translation. We find that various bandit strategies vastly outperform baseline methods like brute-force searching the most challenging topics.

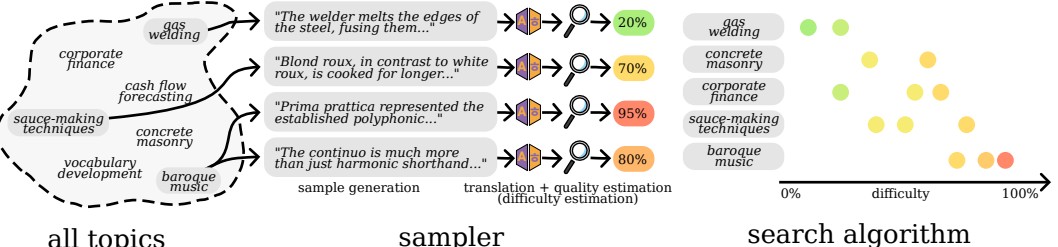

Figure 1: Illustration of our pipeline. Given a large set of all topics, the sampler can draw an example from a topic and estimate its difficulty. The goal of the search algorithm is to find the most difficult topic with as few samplings as possible.

## 1 INTRODUCTION

Effective evaluation is the bedrock of progress in Natural Language Processing, requiring a continuous stream of new test data to challenge model capabilities. While static human labeled benchmarks serve a purpose, they often lack the diversity and sustained difficulty needed to expose the weaknesses of highly capable models, especially for tasks approaching human parity like machine translation (Kocmi et al., 2024; 2025a). While the Internet provides a vast reservoir of complex and varied language, manual curation is non-reproducible, biased, and most importantly infeasible due to the scale and topic diversity.

Inspired by topic modeling (Blei et al., 2003), we conceptualize the massive corpus of Internet data as a structured topic tree, where each text (which can be an input to a model) belongs to a particular topic. Conversely, each topic contains a set of texts. Our objective is to identify the topics that are most challenging for a given model. The core problem lies in the disconnect between a topic's perceived difficulty by a human and its actual difficulty for a model. For example, a topic that is complex for humans, such as Baroque music theory, may be easily processed by any model for a particular task. Conversely, a seemingly unassuming topic like concrete masonry can be challenging if its highly specialized and ambiguous terminology was absent during the model's training data (see

Figure 1). However, identifying the most challenging topics (to fulfill our evaluation desideratum) is not a trivial classification task. The task requires an expensive sample-and-evaluate process. For a given model, each difficulty estimation of a source text involves generating an output and then estimating its quality to determine the difficulty of the source text (Proietti et al., 2025). Applying this process exhaustively multiple times across every potential topic would be computationally intractable. Still, this inefficient, non-systematic sampling is the only current option.

This work, for the example task of machine translation, introduces a budget-constrained algorithm to automatically curate Internet data that reveals weaknesses in targeted models. To navigate the massive search space efficiently, we formalize this discovery process as a multi-armed bandit. In this framework, each topic is an "arm," and pulling an arm corresponds to a single difficulty estimation of that topic. This estimation is a computationally expensive process: it involves sampling a text from the topic, generating a translation with a target model, and evaluating the translation's quality to determine the text's difficulty (see Figure 1). Our goal is to efficiently allocate a fixed budget of these "pulls" to identify the topics that yield the most consistently difficult examples—a task known as best arm identification (Audibert et al., 2010). This approach allows us to strategically explore the vast space of topics, focusing resources on the most promising candidates (most difficult topics) while avoiding blind brute-force evaluation.

We focus on machine translation as a prototypical task, which has suffered from the lack of challenging examples (Proietti et al., 2025) and at the same time any text on the Internet is a possible input that can be translated. To show the aptness and efficiency of our framework we build a search pipeline that collects texts from the Internet.[1] We demonstrate that our bandit-based search vastly outperforms naive sampling strategies, providing an efficient and scalable method for discovering challenging test data for machine translation.

## 2 METHODS

We first describe the preliminaries of difficulty estimation for machine translation, then our framing, the specific search algorithms, and lastly generating data.

**Difficulty estimation.** To estimate the difficulty of a source text $s$, we assess the quality of its translation produced by set of models $m \in M$. The model first generates a translation $o = m(s)$. Subsequently, an error detection model, such as a quality estimation metric trained on human-annotated data, takes the pair $(s, o)$ as input to estimate the number of errors in the translation. These error detection models are either trained based on human-labeled data or LLM-as-a-judge (Freitag et al., 2024). In our case we use GEMBA (Kocmi & Federmann, 2023) based on Gemini-2.5-pro for quality estimation (see prompts in Appendix E). We use the average quality estimation $\frac{\sum_{m \in M} \mathrm{qe}(s, m(s))}{|M|}$ as an inverse proxy for the difficulty of the text $s$. **The difficulty estimation is calculated as 100-QE score.** The approximation of sample difficulty via quality estimation of outputs of a set of models is known as artificial crowd (Zouhar et al., 2025a). Other options to estimate the difficulty are, for example, source-only quality estimation models or LLMs (Proietti et al., 2025), which correlate equally with human judgment. Our framing is independent of the choice of the difficulty estimator.

### 2.1 FINDING DIFFICULT TOPICS AS A MULTI-ARMED BANDIT.

A topic $t$ is a distribution. A sample from a topic $x \sim t$ is a piece of input text and has an associated difficulty score $d_x$. Drawing a single sample $x$ from topic $t$, translating it, and estimating its difficulty has a cost of 1, and $t^\star$ denotes the set of samples that have been drawn from $t$. The difficult topic search task is: given $T = \{t\}_{j=1}^{|T|}$, find top-$k_{t \in T}$ $\mathbb{E}[d_x | x \sim t]$ at example budget $B$, so $\sum_{t \in T} |t^\star| \leq B$. We evaluate any algorithm that selects some $\hat{T} \subseteq T, |\hat{T}| = k$ with budget $B$ by $\frac{\sum_{\hat{t} \in \hat{T}} \mathbb{E}[d_x | x \sim \hat{t}]}{k}$ (i.e. selects the $k$ topics with the highest average difficulty), where higher is better. In the context of machine translation, $t$ is the topic, such as "1990s business news", $x$ is a text in the source language and $d_x$ is the difficulty score.

---

[1]Our formulation is compatible with any NLP task where (1) anything from the Internet is a potentially valid input, and (2) the difficulty of the input can be estimated.

**Bandit**($T$: topics, $B$: budget, $k$):
1: **while** $\sum_{t \in T} |t^\star| < B$
     # Choose domain to sample from.
2:     $t \leftarrow \text{ChooseToSample}(T)$
3:     $x \sim t, t^\star \leftarrow t^\star \cup \{x\}$
     # Select most difficult.
4: **return** $\text{top-}k_{t \in T} \frac{\sum_{x \in t^\star} d_x}{|t^\star|}$

Algorithm 1: General algorithm for non-structured search as a multi-armed bandit. The stopping criterion is given implicitly by reaching the budget. The selection criterion is simply the node with lowest observed maximum. The ChooseToSample functions are instantiated by Algorithms 2 to 4.

**BruteChoose**($T$: topics, $c$: cap):
1: **return** $\text{Uniform}(\{t | t \in T, |t^\star| < c\})$

Algorithm 2: Brute algorithm samples from a random topic limited by cap $c$.

**GreedyChoose**($T$: topics, $c$: cap):
1: **if** $\exists t \in T : |t^\star| = 0$
2:     **return** $\text{Uniform}(\{t | t \in T, |t^\star| = 0\})$
3: **else**
4:     **return** $\arg\max_{t \in T, |t^\star| < c} \frac{\sum_{x \in t^\star} d_x}{|t^\star|}$

Algorithm 3: Greedy algorithm first samples from all topics and then exploits the most difficult one limited by cap $c$.

**EpsilonGreedyChoose**($T$: topics, $c$: cap, $\epsilon$: exploration):
1: **if** $\exists t \in T : |t^\star| = 0 \;\wedge\; \text{Rand}() < \epsilon$
2:     **return** $\text{Uniform}(\{t | t \in T, |t^\star| = 0\})$         # Select a yet non-explored topic.
3: **else**
4:     **return** $\arg\max_{t \in T, |t^\star| < c} \frac{\sum_{x \in t^\star} d_x}{|t^\star|}$         # Exploit the most promising topic.

Algorithm 4: Epsilon-Greedy algorithm stochastically switches between exploitation and exploration. The exploitation is limited by cap $c$.

## 2.2 SEARCH ALGORITHMS

The general form of finding the best $t \in T$ is shown in Algorithm 1. Repeatedly, until we reach a budget, we select a topic to sample from (pull an arm), and at the end select the topic with the highest observed difficulty. For our task, we use a series of increasingly complex selection methods for the topic to sample from. The brute-force in Algorithm 2 is the most basic approach which repeatedly samples from a random topic. At each step a topic with the highest difficulty is selected. This and all other methods have a hyperparameter $c$, which caps the maximum number that we can sample from a single topic to not waste too much of the budget (formally honeypot problem in reinforcement learning). Still, this approach is uninformed and wastes budget even on topics that do not look promising. In contrast, Algorithm 3 exploits and selects the topic that has currently the highest observed difficulty. This exploitation, however, requires all topics to be sampled at least once in order to commence.

Scoring all topics even once can be prohibitively expensive. To alleviate this, we need to be able to reliably start scoring and exploiting topics even before all of them are sampled from. For this reason, we use $\epsilon$-greedy algorithm which stochastically switches between exploring never-sampled topics and exploiting (see Algorithm 4).

We also consider a batched version of these algorithms by replacing steps 2 and 3 in Algorithm 1 with choosing top-$b$ topics at once. This helps avoid local minima and through regularization.

## 2.3 GENERATING AND SAMPLING FROM INTERNET DATA

We first generate $T$ hierarchically by starting with top-level topics of interest: "science," "business," "law," "education," and "culture." Then, we recursively specialize each topic, so "business" creates "finance," "business innovation," "globalization", and specializing "finance" then creates "corporate finance" and so on. For each of the topics we generate five expanded subtopics and repeat this five times, which yields $|T| = 3.2k$. This process is done with a prompted LLM (see Appendix E for details) and is generally inexpensive. This process can also be replaced by using a predefined taxonomy, a list of topics, other taxonomies covering seeds for data, or even Internet domains.

In our setting, drawing a sample $x$ from a topic $t$ is achieved through the auto-regressive generation of the language model. Specifically, we enable Google search tool calling[2] when asking the LLM

---

[2]gemini-api/docs/google-search, accessed 07-2025.

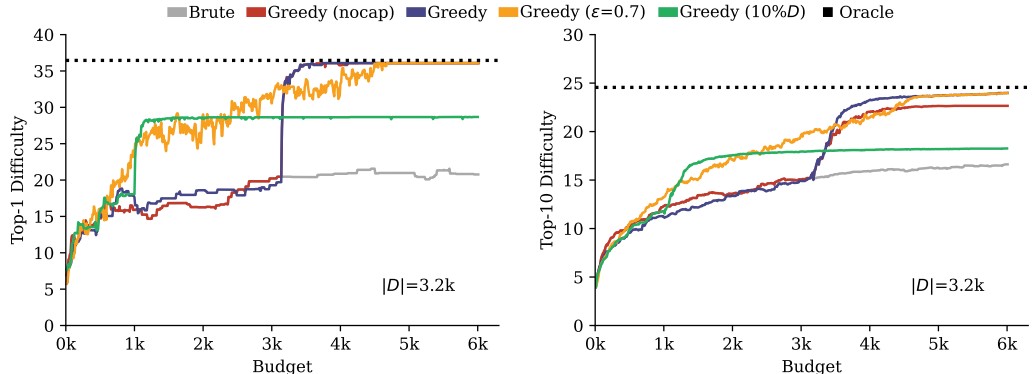

Figure 2: Results for algorithms measured with top-1 and top-10 difficulty. All algorithms have the same budget and the cost of a single sampling is 1.

to generate text based on topic $t$. For a given prompt of "find all relevant topics about News in English," the LLM will first make Google Search call to extract all relevent snippets about "news" and add them into the context of LLM's input. Then based on all the search returned context, the LLM will extract texts are the most relevant to the topic. To make sure the LLM generates texts based on the real texts, we request the LLM to pair texts with given URL source for provenance.

## 3   EXPERIMENTS

We first detail and compare the proposed methods for searching for difficult topics. Then, we describe the topics that were found by our pipeline in comparison to existing datasets, which is our main finding.

**Setup.**   We study searching for sources in English and translating them into Czech, Chinese, German, and Ukrainian using Google Translate,[3] Gemini 2.5 Pro (Gemini Team et al., 2025), and Gemma 3 (Gemma Team et al., 2025). This selection covers three language families (Germanic, Slavic, Sinitic), high- and low- language data resourcefulness, and diverse machine translation models. The goal is to efficiently find top-1 or top-10 most difficult topics within the $T$ created by the aforementioned setup.

### 3.1   COMPARISON OF ALGORITHMS

We evaluate the search algorithms based on the difficulty of the final topic they choose for a particular budget. In Figure 2 we compare oracle (top-$k_{t \in T} \frac{\sum_{x \in t} d_x}{|t|}$), uninformed brute search (Algorithm 2), greedy search (Algorithm 3), greedy search with no exploitation constraints, epsilon-greedy search (Algorithm 4), and greedy search on randomly sampled 10% of the data. The results show that the brute search is near unusable and that exploitation is needed. While the greedy approach generally works, it requires at least one sampling of each topic and only then it can steeply exploit. Even this might be prohibitively expensive, which is why we include the epsilon-greedy algorithm which begins exploiting early. This outperforms running the greedy algorithm on a subset which can lose on more difficult topics by chance. Lastly, the unconstrained version of greedy search ($c = \infty$) might get stuck exploiting a locally optimal topic, which shows that a cap on the number of exploits is desirable.

Overall, the topic set in Figure 2 contained $\sim$3000 topics and the best performing algorithms selected near-oracle topics (absolute difference in difficulty $\Delta < 0.1$) by 4500 steps, which is less than two samples per topic. The topic set was pre-generated with 25 samples per each topic, so this corresponds to 6% of the cost of sampling each topic the maximum number of times. In Appendix C we discuss other common search algorithms and why they might not be suitable to the current task.

---

[3]translate.google.com, accessed 07-2025.

| Existing domains | | Difficulty↑ | Words↓ | Example |
|---|---|---|---|---|
| **WMT 2024** Kocmi et al., 2024 | Social | 10.1 | 16 | In general I really like the interplay between the two games. The advantage of shortform stories is that you can "skip to the good part"... |
| | Literary | 8.3 | 38 | The advancement of Humanity never ceased, even for a moment–during difficult times we grow and adapt once again. The cities are as prosperous as ever, and our technological advancement is rising. ... |
| | Speech | 8.3 | 73 | Cheers, y'all. Now check it out. I really didn't even eat enough to be wiping my mouth, but I can tell you this, my mouth is salivating though. .... |
| | News | 6.4 | 54 | "People Swimming in the Swimming Pool" from 2022 is one Vicente Siso artwork that will display at Tierra del Sol Gallery beginning Jan. 13. (photo courtesy of Vicente Siso)... |
| **WMT 2025** Kocmi et al., 2025a | Speech | 17.3 | 145 | Gotta watch a netflix show you feel me, but let me know down below. What show should i watch on netflix though? Because i'm i'm really having some trouble to find what show should ... |
| | News | 14.3 | 95 | Some folks really do deserve a badge of honour for their pedantry (C8). Veronica Coyne of Springfield claims that "when bemoaning the loss of the express lane at Woolies "12 items or less,"... |
| | Social | 11.8 | 98 | Another fine evening (ok not really, it's wet and drizzly, but) to continue exploring my stash of Rum from Japan Cor Cor again - this time the "Industrial" .... |
| | Literary | 9.9 | 117 | It had been a remarkable twenty-year pro career, one that most players could only dream of. He wore a gleaming championship ring, a testament to his hard work and dedication .... |
| | Dialogue | 5.7 | 179 | X: I am looking for a cheap hotel with free parking near Cambridge. Y: I have multiple cheap hotels with free parking. What part of town are you interested in staying in?... |
| **FLORES-101** NLLB Team et al., 2022 | Wikinews/disasters and accidents | 4.7 | 18 | At 1:15 a.m. Saturday, according to witnesses, the bus was going through a green light when the car made a turn in front of it.... |
| | Wikivoyage/travel | 4.6 | 21 | Cold weather is perhaps the only real danger the unprepared will face.... |
| | Wikinews/politics | 3.8 | 21 | Mr Costello said that when nuclear power generation becomes economically viable, Australia should pursue its use.... |
| | Wikinews/sports | 3.6 | 18 | Mr Reid managed to drive the New Zealand's A1GP car, Black Beauty at speeds over 160km/h seven times over the bridge.... |

| Our topics | Difficulty↑ | Words↓ | Example |
|---|---|---|---|
| Incarceration Prison vs Jail | 39.5 | 29 | Jails are short-term facilities for temporary detention, ... Prisons are long-term facilities for extended incarceration.... |
| Leasehold Estates Tenancy for Years Periodic Tenancy | 29.6 | 32 | Periodic Tenancy: A non-freehold estate that lasts only from period to period without having any definite duration that is longer than one .... |
| Future Interests Reversions Remainders Executory Interests | 29.5 | 34 | Future interests are legal rights to property ownership that may become possessory later. They arise when a grantor conveys.... |
| Removal Jurisdiction State to Federal Court | 21.2 | 30 | For removal based on diversity jurisdiction, the amount in controversy must exceed $75,000, and there must be complete diversity of .... |
| Victim Impact Statements Role | 21.0 | 34 | Victim impact statements detail the emotional, physical, and financial consequences of a crime. They can be written or oral and are... |

Table 1: Comparison of topics from existing dataset (top) and topics found by our $\epsilon$-greedy algorithm (bottom). All topics that are found by our algorithm are more difficult compared to existing topics despite having lower average number of words (difficulty scales with length). Appendix Table 5 illustrates that our most challenging topic, "Incarceration: Prison vs. Jail," is comparably difficult to most challenging subsets of existing benchmarks. We include detailed case study of challenge source texts and model mistakes in Appendix Tables 6 to 8.

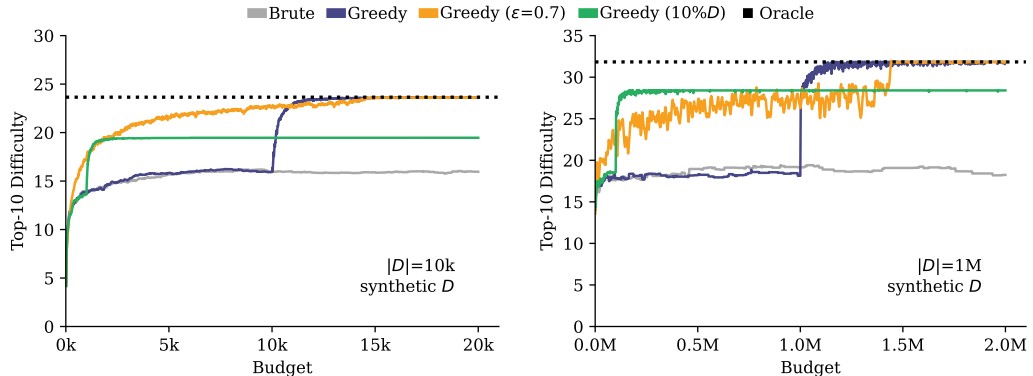

Figure 3: Results for algorithms measured with top-10 difficulty on synthetically large $T$. All algorithms have the same budget and the cost of a single difficult estimation is 1.

## 3.2 DISCOVERED DIFFICULT TOPICS

In this section we compare the difficulty of the discovered topics with existing benchmarks. Direct comparison of difficulty is not straightforward because text length naturally increases chance of errors. While we do control for the sample length (samples in all topics are 20 to 40 words), we can not match to a predefined length of other testsets which each have different average sample lengths.

We compare to the popular machine translation testsets: WMT 2024 (Kocmi et al., 2024), WMT 2025 (Kocmi et al., 2025b), and FLORES-101 (NLLB Team et al., 2022) in Table 1 (top). Our found topics are computed with respect to the average difficulty across all languages and models in Table 1 (bottom). While direct comparison is not possible given the granularity, sample length, and sample count differences, the oracle topics also found by the search algorithm have comparable difficulty (number of errors) while being generally much shorter. Having more challenging test set is beneficial for spotting and improving model failures but also for better benchmarking, which we discuss in Appendix B. Appendix Table 5 illustrates that our most challenging topic, "Incarceration: Prison vs. Jail," achieves a difficulty comparable to that of the most challenging subsets identified within these three benchmarks. Furthermore, all five of the most challenging topics discovered by our algorithm consistently demonstrate a higher difficulty level than most subsets across these benchmarks.

We note a potential limitation of our approach: our framework seeks out outlier topics that are the most difficult. This difficulty is with respect to some estimator (Section 2) and it is possible that the search yields topics that are outlier only due to the estimator's noise. However, we treat the output of the estimator as the ground truth and admit only stochasticity in topic difficulty distribution and example difficulty distribution within a particular topic. The difficulty estimator will benefit from future improvements in quality and difficulty estimators and could even be replaced with more accurate human-in-the-loop.

## 3.3 SEARCH ALGORITHMS AT SCALE

In this section we verify that scaling the size of $T$ still leads to retrieving more difficult topics. We do so by synthesizing arbitrarily large $T$. For this, we need to be able to estimate $d_x$ based on the true distribution. We model $d_x$ with a generative process: First, we sample $\mu_t$, the mean of a topic $t$ from an empirically fitted Gaussian mixtrue distribution (Figure 4). Then, we sample from the distribution of $d_x$ conditioned on $\mu_t$: $\mathcal{N}(\mu_t, \sigma^2)$ where $\sigma^2$ is estimated from true data. For a synthetic topic set of size $|T_S|$ we first generate $|T_S|$ means and then sample $d_x$ conditioned on those means.

In Figure 3 we show search algorithms on these synthetic topic sets of sizes 10k and 1M. The search algorithms are able to reach close to oracle difficulty with approximately 1.5 samples per topic ($\sim$6%). Naturally, with a larger pool of topics, we would expect to obtain a higher top-$k$ $\mathbb{E}[d_x|x \sim t]$. In Figure 5 we show that with increasing the topic set, we also increase the top difficulty. The relationship seems to be similar to logarithmic at our scale (up to 10M): a 10-fold increase in topic set size leads to +5 difficulty. This can also be confirmed formally as the maximum of $|T|$ samples from a normal distribution is asymptotically $\sqrt{\log |T|}$ (Leadbetter et al., 1983).

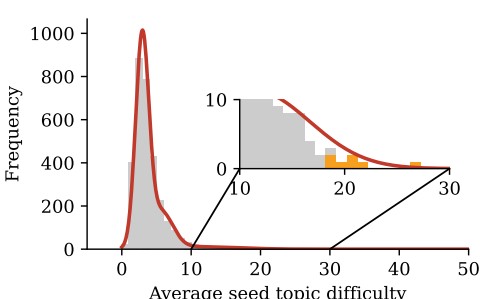

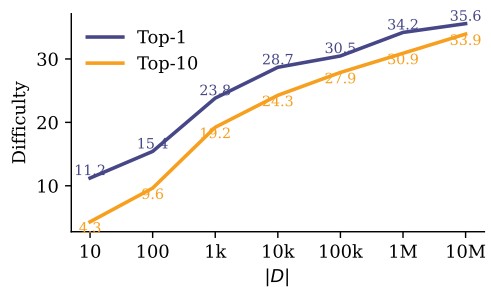

Figure 4: Distribution of topic difficulty (gray), best fit for Gaussian mixture model distribution with 3 components (red) used for synthetic scaling, and top-10 empirical oracle (orange).

Figure 5: Estimated oracle difficulty for topic sizes using a sample generative process as in Figure 4. The synthetic generation is more conservative than the real data (at $|T| = 3.2k$ the top-1 is 36 and top-10 is 25 from Figure 2).

| Error Severity | | Error Category | | Error Type | |
|---|---|---|---|---|---|
| Major | 70.7% | Terminology | 47.7% | Inappropriate for Context | 47.6% |
| Minor | 21.3% | Accuracy | 42.3% | Mistranslation | 35.9% |
| Critical | 8.0% | Style | 5.1% | Omission | 5.6% |
| | | Fluency | 4.9% | Awkward | 5.5% |
| | | | | Untranslated | 3.6% |
| | | | | Grammar | 1.8% |

Table 2: Distributions of AutoMQM (Fernandes et al., 2023) error severities, categories, and types for our collected test set of 250 examples using an $\epsilon$-greedy algorithm (averaged across four language directions and three models). The test set primarily induces accuracy and terminology errors, resulting in mistranslations and contextually inappropriate outputs. Most of these errors are classified as major (altering the meaning).

### 3.4 ERROR TYPE ANALYSIS

To thoroughly analyze translation quality, we use AutoMQM (Fernandes et al., 2023) to obtain detailed error categories, types, and severities for our collected test set on three translation models: Gemini-2.5-Pro, Gemma3-27b and Google translate across four language directions.[4] The leftmost part in Table 2 shows that most of these errors are classified as major errors (changing the meaning of the sentence).

As depicted in Table 2, the test set, generated using an $\epsilon$-greedy algorithm, primarily exposes models to accuracy and terminology errors. These manifest as mistranslations and contextually inappropriate outputs. This outcome, however, does not imply that our pipeline is limited to finding errors solely related to terminology or accuracy. Our difficulty estimation model guides the search process towards generally challenging examples for the studied translation models, rather than focusing on specific error types. Consequently, the observed error distributions naturally reflect the inherent weaknesses of these models. The framework is independent of the specific difficulty estimation. For example, the quality estimation model could be replaced with a fluency metric to guide the search process specifically towards identifying fluency errors. Appendix Tables 6 to 8 presents examples of major terminology and accuracy errors, primarily focusing on terms that were mistranslated due to a lack of contextual understanding or unrecognized terminologies by the models.

### 3.5 COST ANALYSIS

In Table 3, we present the top-10 difficulty achieved at various monetary costs. Our epsilon-greedy search consistently demonstrates a substantial advantage over uninformed brute search in identifying texts with high difficulty. Specifically, epsilon-greedy achieves a much higher level of difficulty for

---

[4]AutoMQM is an LLM-based pipeline to identify and classify specific errors in translated texts according to the Multidimensional Quality Metrics (MQM, Freitag et al., 2021).

| # of Requests | Cost $ | | | | Top-10 Difficulty | |
|---|---|---|---|---|---|---|
| | **Search** | **Translation** | **QE** | **Total** | **Brute** | **Greedy ($\epsilon$=0.7)** |
| 20k | $87 | $2 | $15 | $104 | 16.3 | 19.7 |
| 200k | $867 | $21 | $152 | $1040 | 17.9 | 25.3 |
| 2M | $8668 | $215 | $1520 | $10403 | 18.2 | 31.8 |

Table 3: At different monetary costs, our epsilon-greedy search offers an advantage in identifying topics with high difficulty over uninformed brute search. Notably, for a mere $104, epsilon-greedy achieves a level of difficulty that brute search cannot attain, even with an investment of $10,403.

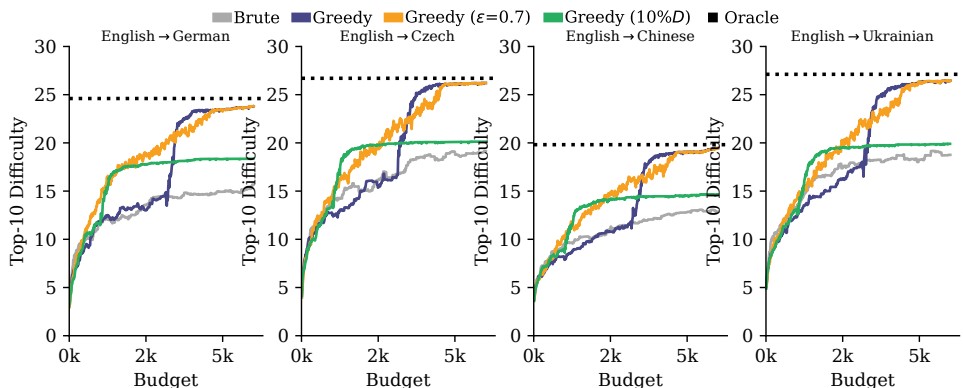

Figure 6: Our algorithms performed consistently across all languages directions.

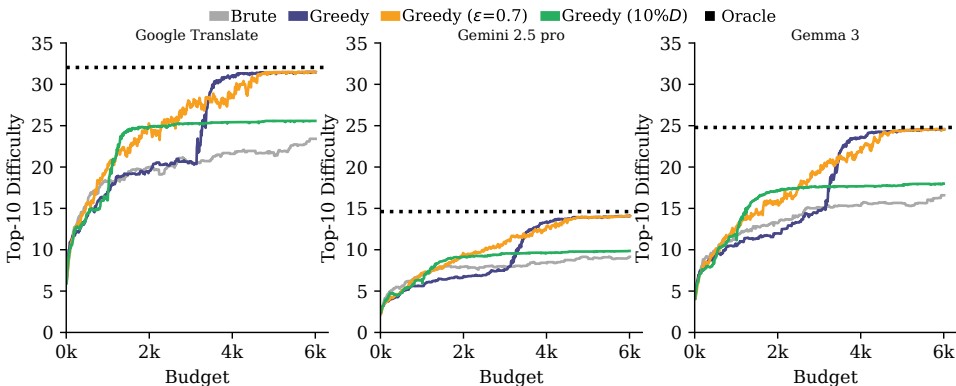

Figure 7: Our algorithms can consistently identify challenge source texts across Gemini-2.5-pro, Gemma3-27B and Google Translate.

a mere $104. In stark contrast, brute search requires an investment of approximately $10,403 to perform 2 million search requests, yet it fails to reach the same level of difficulty achieved by our method. Due to real budgetary constraints during the preparation of this paper, the performance for extensive search requests (e.g., up to 2 million) presented in Table 3 is estimated. These estimations are based on extrapolations from synthetic data, as detailed in Figures 4 and 5.

## 3.6 PER-LANGUAGE AND PER-MODEL ANALYSIS

As shown in Figure 6, our best search method, epsilon-greedy, consistently identifies near-optimal challenge samples across all four languages tested. The relative performance of the search algorithms remains stable across these languages. Notably, epsilon-greedy successfully finds difficult samples (around a score of 20) even for high-resource language pairs, like English→Chinese. The Figure 7 also shows the robustness of our algorithms, as they consistently find near-optimal chal-

---

[3]The sampling and quality estimation use Gemini-2.5-pro ($1.25/1M input and $10.00/1M output tokens).

| | Is $k$th of.. | | | | | Is k-th of.. | | |
|---|---|---|---|---|---|---|---|---|
| Top-10 of.. | **Czech** | **Chinese** | **German** | **Ukrainian** | Top-10.. | **Gemini** | **Gemma** | **G.Trans.** |
| **Czech** | 5.5 | 82.5 | 47.4 | 55.9 | **Gemini** | 5.5 | 30.8 | 10.3 |
| **Chinese** | 190.3 | 5.5 | 313.7 | 100.0 | **Gemma** | 48.4 | 5.5 | 47.3 |
| **German** | 136.3 | 345.4 | 5.5 | 108.8 | **G.Trans.** | 16.8 | 20.9 | 5.5 |
| **Ukrainian** | 20.5 | 37.6 | 63.5 | 5.5 | | | | |

Table 4: Cross-language and -model topic difficulty analysis. Difficult topics are largely difficult across languages and models.

lenge samples for different models. More importantly, epsilon-greedy-greedy proves effective even against a top-performing translation LLM, Gemini-2.5-pro.

Finally, Table 4 reveals that the most challenging topics are dependent on both the specific language and the model. The left side of the table, for instance, shows a cross-difficulty relationship between the related languages Czech and Ukrainian: topics challenging for Czech tend to be relatively difficult for Ukrainian, and vice versa.

## 4 RELATED WORK

**Difficult examples.** The example difficulty is tied to evaluating the example-level quality of model outputs. In machine translation, this is commonly done with reference-free (i.e. no ground truth) automated metrics, such as SEScore (Xu et al., 2022; 2023a), InstructScore (Xu et al., 2023b), COMET (Rei et al., 2020) or MetricX (Juraska et al., 2023), which provide a score corresponding to the output quality. This score can be used as a proxy for difficulty. Proietti et al. (2025) train a model that predicts the expected model performance based on just the source, which can be used for difficulty estimation. Knowing the difficulty of an example for models has many uses, ranging from curriculum learning (Jia et al., 2025) to more efficient evaluation (Zhan et al., 2021). Approaches for searching for difficult examples are often limited to some apriori knowledge of difficulty, which guides the selection of syntactically complex texts or texts with rare words (Chen et al., 2023). A different approach generates, not searches, for difficult-to-translate texts (Pombal et al., 2025; Lu et al., 2025; Zouhar et al., 2025b) or otherwise adversarial examples (Zhang et al., 2021; Sadrizadeh et al., 2024).

**Bandits.** Our task setup in Section 2 corresponds to the Best Arm Identification problem (Audibert et al., 2010). Many works also make use of some features of the arms, such as their similarities, to inform the choices (Li et al., 2010). In machine translation, Cheng et al. (2025); Zouhar et al. (2025c) use a generalization of the multi-armed bandit to improve machine translation and quality estimation efficiency. Also in machine translation, Kumar et al. (2019); Kreutzer et al. (2021) use bandit formulation for training data selection and curriculum learning.

## 5 CONCLUSION

To advance Natural Language Processing, future efforts must target challenging "tail" examples that still present headroom for improvement. Efficiently identifying this tail at scale is not straightforward, leading us to frame the task as a best multi-armed bandit arm identification problem. Each topic represents an arm, which can be "pulled" at a cost to sample from it and estimate its difficulty. With an epsilon-greedy exploration-exploitation strategy, we efficiently identify near-oracle topics. The topics automatically discovered for machine translation yield texts with difficulty levels surpassing standard benchmarks like WMT and FLORES. This framing facilitates a transition from generic static benchmarks to dynamically collected ones, driven by a difficulty estimator, such as artificial crowd with a quality estimator. Future work should extend this approach to other NLP tasks or integrate it into an active learning setup where examples are used, for example, for online learning rather than solely for evaluation.

# 6 REPRODUCIBILITY STATEMENT

All pseudocode for the search algorithms is provided in the main paper and the appendix (Algorithms 1 to 4 and 5). The specific prompts used for hierarchical topic generation, grounded text sampling from the Internet, translation, and quality estimation are detailed in Appendix E. Key hyperparameters for our search algorithms are specified in the experiments section (e.g., $\epsilon=0.7$ for the epsilon-greedy algorithm in Figure 2). The experiments rely on publicly available translation models (Gemma 3) and commercial APIs (Google Translate, Gemini 2.5 Pro), for which we provide a timestamp in Section 3.

A detailed breakdown of the computational costs, which extends Section 3.5, is available in Appendix D. A key consideration for grounded text sampling from the Internet is that the availability of relevant online content can restrict the ability to find 25 distinct texts for all topics, e.g. because of their relative obscurity. Consequently, 10% of our topics were discarded in this study because the target sentence count could not be achieved (therefore the empirical dataset has 3.2k topics and not $\sum_1^5 5^n \approx 3.9$k topics).

We plan to release the code and the curated dataset of difficult topics upon publication to facilitate further research. We used Gemini-2.5-Pro to polish the writing of this paper.

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

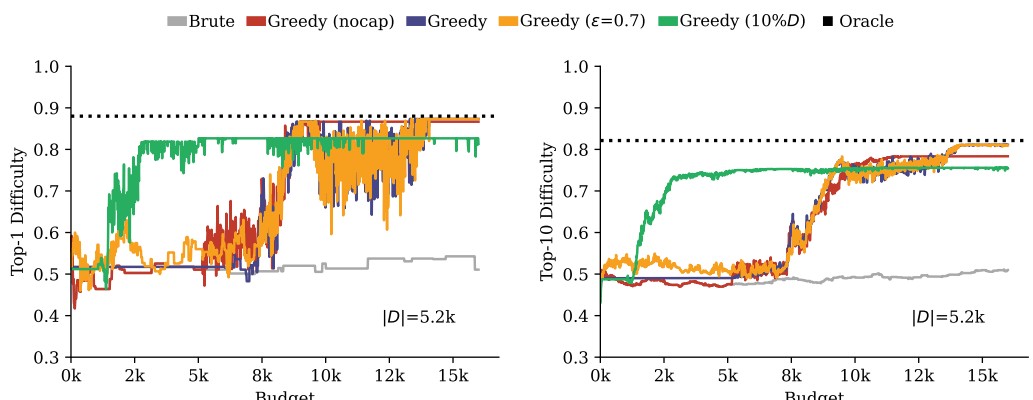

Figure 8: Results for algorithms measured with top-1 and top-10 difficulty at knowledge question answering task. The score range for each domain is between $0$ to $1$. The difficulty esitmation is measured as $1$ - quality estimation. All algorithms have the same budget and the cost of a single sampling is $1$.

## A KNOWLEDGE QUESTION ANSWERING

To demonstrate the generalization capabilities of our proposed pipeline, we extend our framework to knowledge question answering, following the approach of ECLeKTic (Goldman et al., 2025). Analogous to our machine translation pipeline, we use the prompt in Appendix E to generate $6,000$ domains, each containing 25 distinct topics. For each topic, we employ a search agent to formulate queries and retrieve the most relevant Google snippets. Conditioned on this context, we instruct the model—using the prompt in Appendix E—to synthesize factual question-response pairs. Inspired by ECLeKTic, we target factual questions with definitive answers. Crucially, while the model relies on the retrieved snippets to generate the pairs, we explicitly instruct it to formulate questions that can be answered without access to the context. Finally, to ensure data consistency, we filter out any domain containing at least one topic where the questions remain context-dependent. Ultimately, we obtain a dataset of $5,160$ domains, each containing 25 subtopics.

Since the question-response pairs are synthesized directly from source snippets, we treat them as ground truth and employ reference-based evaluation for the model-generated answers (see Appendix E for specific prompts). Autorater scale range is between $0$ to $1$. Therefore, difficult estimation is $1$ - quality estimation. We include three model to answer factual questions: 1) Gemini without thinking. We set the thinking budget to $0$ when sampling answers. 2) Gemini with thinking. We use the default thinking budget when sampling answers. 3) Gemini with search turned on. The final quality estimation is calculated by averaging the performance across the responses from these three models.

In Figure 8, we observe that Knowledge QA trends align closely with the translation task results. Bandit-based algorithms greatly outperform the brute-force approach, further demonstrating the generalizability of our framework. Greedy and $\epsilon$-Greedy surpass other bandit methods, demonstrating the value of exploitation. Although Greedy (10%) excels at rapidly locating sub-optimal domains in early steps, $\epsilon$-Greedy achieves higher performance with a larger budget. Most importantly, we recover the oracle difficult samples by exploring only 10% of the total search space ($13k$ out of $130k$ subtopics)—equating to fewer than 3 samples per domain.

## B DIFFICULT TEXTS RANK MODELS BETTER

Having a difficult testset at hand is beneficial for many reasons. One of them is the higher discriminability; i.e. the ability of this data in distinguishing good from bad models and ranking them. Zouhar et al. (2025a) find that difficult examples and examples where models have varying scores (high variance) tend to be better at efficiently ranking the models. In this section, we confirm this for our method of obtaining difficult data.

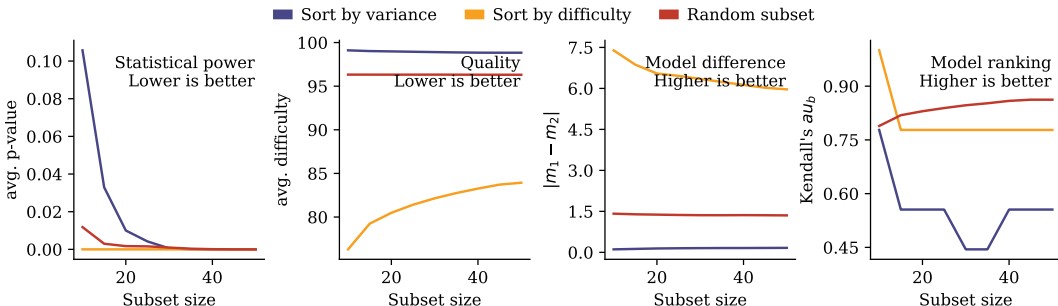

Figure 9: Various measures of subset utility: statistical power (p-value between adjacent models), average difficulty, average difference between adjacent models, ranking similarity between a subset and the whole testset (gold).

In Figure 9 we measure 4 key properties of a challenge set: (1) statistical discriminability, (2) average difficulty, (3) differences in average model scores, and (4) ranking on the subset with respect to the ranking on the whole set (proxy for gold model ranking). Systematically, the difficult challenge set, based on our $\epsilon$-greedy selection strategy outperforms random test example selection. We also include a challenge with the highest variance between models, as suggested by Zouhar et al. (2025a), though this does not perform consistently well across all criteria.

## C  OTHER SEARCH ALGORITHMS

### C.1  CONTEXTUAL BANDIT

Contextual bandit makes an observation that topics that are similar to each other are likely going to have similar sample difficulty. Therefore, even if a topic has never been sampled from, but the difficulty of all its neighbours is low, we do not have to spend budget on sampling from it. This is shown in Figure 10: The topic with difficult neighbours is prioritized over the topics with easy neighbours which are never going to be sampled from. This extension of the greedy approach is known as contextual bandit where we can consider some features of the arms (in our case topics) for the selection. Practically, we first make sure that all topics are scoreable (i.e. they have been sampled from or have at least two neighbours with samples). Then, we interpolate between the topic's score and the scores of its neighbours based on the similarity. For computing the similarity, we use the overlap in keywords (Jaccard index).

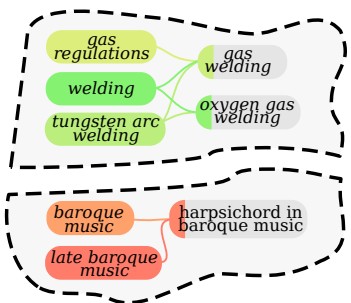

Figure 10: Illustration of neighbour effect on the selection of topics to sample from. Even though *gas welding* and *oxygen gas welding* topics were never sampled from, the neighbours suggest that those topics will not be difficult. In contrast, *harpsichord in baroque music* will likely be difficult.

### C.2  UPPER CONFIDENCE BOUND BANDIT

While there are many algorithms for multi-armed bandit, the biggest obstacle in our case in Figure 2 is the cold-start and initialization phase needed (before all topics have been exploited at least once) for the greedy search. Improvements that take into account the confidence, such as upper confidence bound bandit (Auer et al., 2002), would still require sampling of all initial topics. Their advantage would only be faster descent after the initialization phase, which is already steep for our method in comparison to the cost of the initialization phase.

## D  COST ANALYSIS

For data generation, the average prompt length was 146 input tokens, resulting in an average generated source output of 94 tokens. Additional grounded search costs amounted to $35 per 1,000

**ContextualChoose**($T$: topics, $c$: cap):
1: **if** $\exists t \in T : \text{ContextualScore}(t) = ⑦$
2:     **return** $\text{uniform}(\{t | t \in T, |t^\star| = 0\})$
3: **else**
4:     **return** $\arg\max_{t \in T, |t^\star| < c} \text{ContextualScore}(t)$

**ContextualScore**($t$: topic):
1: $N \leftarrow \{t_o | t_o \in T \setminus \{t\}, \text{sim}(t, t_o) > 0 \wedge |t_o^\star| > 0\}$      # Find all scoreable neighbours
2: **if** $|t^\star| = 0 \wedge |N| < 2$:      # If this topic can not be scored, return ⑦
3:     **return** ⑦
4: **else**
5:     $\beta = \begin{cases} 0 & \text{if } |t^\star| = 0 \\ 0.5 & \text{if } |t^\star| = 1 \\ 1 & \text{if } |t^\star| \geq 2 \end{cases}$      # Interpolate between own score and score by neighbours
6:     **return** $\beta \cdot \frac{\sum_{x \in t^\star} d_x}{|t^\star|} + (1 - \beta) \cdot \text{softmax}(\langle \text{sim}(t, t_o) | t_o \in N \rangle) \cdot \langle \frac{\sum_{x \in t_o} d_x}{|t_o^\star|} | t_o \in N \rangle$

Algorithm 5: Contextual bandit algorithm first makes all seed topics scoreable and then exploits the most difficult one limited by cap $c$.

requests. To collect 25 source sentences per topic, the Google Search-grounded Gemini model was prompted to extract relevant sentences from its search snippets, typically requiring an average of three queries per topic. For translation, the average prompt and output lengths were 107 and 76 tokens, respectively. Quality estimation prompts and outputs averaged 310 and 595 tokens, respectively.

# E PROMPTS

**Topic generation.**   You are an expert ontologist specializing in domain_name. Your
task is to generate a comprehensive concept tree for this domain. Please adhere
to the following specifications:
Output Format: Generate a single Python code block containing a nested
dictionary representing the concept tree.

Tree Structure:
Root Node: The root of the tree must be 'domain_name'.
Depth: The tree must have a depth of number_of_levels, meaning there are
number_of_levels levels of subtopics beneath the root node.
Branching Factor: Each parent node (non-leaf node) must generate exactly
branching_factor unique child nodes (subtopics).
Node Naming (Crucial):
Self-Contained: Each node name (the dictionary key) must be a self-contained
and specific phrase suitable for a direct Google search. It must be fully
understandable without knowing its parent topic.
Language: All node names must be in language.
Example Structure: The final output should follow this nested dictionary format
(with no additional comments or text):

```
"[Root Node Name]":  {
   "[Subtopic 1.1]":  {
     "[Subtopic 1.1.1]":  {
       # ...  continue for specified depth
     },
     "[Subtopic 1.1.2]":  { ...  },
     "[Subtopic 1.1.3]":  { ...  },
     "[Subtopic 1.1.4]":  { ...  }
   },
```

```
     "[Subtopic 1.2]":  { ...  },
     "[Subtopic 1.3]":  { ...  },
     "[Subtopic 1.4]":  { ...  }

 }
```

**Sampling source text from topic.**    Please use Google search to find all relevant topics about SEARCH_KEY_WORDS in LANG. Then extract all relevant snippet contents in the format of JSON. Each extracted content should be approximately 20-40 words and distinct from each other.  Please make sure to extract all relevant contents.

```
{
"extracted_snippets":  [
  {
    "text":   "content 1",
    "source_url":  "http://example.com/source_1"
  },
  ...
  {
    "text":   "content n",
    "source_url":  "http://example.com/source_n"
  },
  ]
}
```

**Quality Estimation.**    Evaluate the quality of the translation on a scale from 0 to 100.  Roughly:
100 – Perfect
95 – Excellent (closely aligned with the source)
80 – Very good (minor style choice)
60 – Fair (some inaccuracies or fluency errors)
40 – Poor (multiple inaccuracies or fluency errors)
0 – Inadequate (unrelated, completely wrong)
First, think about all the errors in the translation and their severity (very briefly, max few words per error).  At the end, output a single line in the format like as follows:
SCORE |||70.8|||
The last line is important because it will be matched with a regex, so make sure to use the |.
Don't think for too long (max 10 sentences).

SOURCE: |||src|||
TRANSLATION: |||tgt|||

**Translation.**    You are a professional translator.  You are given a source text in src_lang.  You need to translate the source text to tgt_lang.  Don't include any other text except the translation.  Please output the translation between <START OF TRANSLATION> and </END OF TRANSLATION>.  Source text:  src_txt

**Sampling question from topic.**    Task:  Formulate a question in {tar_lang} that requires a deep understanding of a given {src_lang} paragraph.  Requirements:
* Context-Specific:  The question must be answerable solely through information presented within the paragraph, excluding general knowledge or common sense.  * Self-Contained:  The question should be completely self-explanatory, providing all necessary context within its phrasing.  Assume the reader has no access to the paragraph when answering the question.  * Single Concrete Factual Detail:
– The question should not require multiple answers or involve listing multiple details.  – Avoid asking about opinions, interpretations.  – In you can't answer the question, prefer to generate another question.  – Focus on extracting a specific, concrete, factual detail that the paragraph directly states.  – Be specific:  – If you are asking about an entity be clear about it – Use full names for example.  – Mention expected granularity:  If you are asking about a date, instead of asking "when", ask for a decade, year, month, date etc.  If you are

asking about a location, instead of asking "where", ask for a country, state, city, street, landmark etc. – Avoid asking questions that their answers are acronyms. – When formulating a question, you should assume that the person who answers this question does not have access to the paragraph. Avoid using phrases like "According to the paragraph". – Make sure the answer is very short. Even for non-English examples keep the convention of using the special words like "paragraph", "response", "[BEGIN OF QUESTION]", "[END OF QUESTION]", "[BEGIN OF ANSWER]", "[END OF ANSWER]" for specifying the parts being generated. Generate only the question and answer. No need to continue with additional examples. Examples: Paragraph: The Great Barrier Reef is the world's largest coral reef system, composed of over 2,900 individual reefs and 900 islands stretching for over 2,300 kilometers (1,400 mi) over an area of approximately 344,400 square kilometers (133,000 sq mi). The reef is located in the Coral Sea, off the coast of Queensland, Australia. The Great Barrier Reef can be seen from outer space and is the world's biggest single structure made by living organisms. Response: [BEGIN OF QUESTION] Where is the Great Barrier Reef located?[END OF QUESTION] [BEGIN OF ANSWER] Coral Sea, off the coast of Queensland, Australia[END OF ANSWER] Paragraph: Die Cazoo Snookerweltmeisterschaft 2023 wurde vom 15. April bis 1. Mai im Crucible Theatre in Sheffield ausgetragen. Mit ihr endete die Saison 2022/23 der World Snooker Tour.[1] Titelverteidiger Ronnie O'Sullivan scheiterte im Viertelfinale gegen Luca Brecel. Der Belgier erreichte das Finale und schlug dort den vierfachen Weltmeister Mark Selby mit 18:15. Brecel ist damit der erste Kontinentaleuropäer, der Weltmeister wurde. In diesem Jahr wurden noch weitere Bestmarken in Bezug auf die 47-jährige „Crucible-Ära" aufgestellt. Unter anderem übertraf Ronnie O'Sullivan mit seiner 31. Endrundenteilnahme die 30 Teilnahmen von Steve Davis.[2] O'Sullivan erzielte auch sein 200. WM-Century-Break. Zweimal wurde ein Maximum Break erzielt, was es 2008 bereits einmal gegeben hatte; das „perfekte Break" in einem WM-Finale gelang 2023 erstmals Mark Selby. Response: [BEGIN OF QUESTION] Gegen wen verlor Ronnie O'Sullivan im Viertelfinale der Snooker-Weltmeisterschaft 2023?[END OF QUESTION] [BEGIN OF ANSWER] Luca Brecel[END OF ANSWER] Paragraph: context Response:

**QA prompt.** You are an expert in knowledge question answering. Question: {question} Please answer the question between [BEGIN OF ANSWER] and [END OF ANSWER]. Make sure the answer is very short.

**QA Autorater Prompt.** You are an expert evaluator for Knowledge Question Answering. You will be provided with a Question, Context, Reference Answer, and a Model Response. Your goal is to judge the factual accuracy of the Model Response.

Evaluation Criteria: Single Definitive Answer: If the question has only one correct answer, determine if the Model Response creates a semantic match with the Reference Answer. Multiple Definitive Answers: If the question allows for various valid answers, assess factual correctness by verifying answer against the provided Context and the Reference Answer.

Answer 'yes' if the Model Response contains a factual mistake. Answer 'no' if the Model Response is factually correct. Answer 'yes' or 'no' only. At the end, output a single line in the format like as follows: `ANSWER |||yes|||` or `ANSWER |||no|||`

QUESTION: |||question||| CONTEXT: |||context||| REFERENCE ANSWER: ||||{reference_answer}|||| MODEL RESPONSE: ||||{model_response}||||

| Existing domains | | Difficulty↑ | Words↓ | Example |
|---|---|---|---|---|
| WMT 2024 Kocmi et al., 2024 | News | 14.1 | 58 | "People Swimming in the Swimming Pool" from 2022 is one Vicente Siso artwork that will display at Tierra del Sol Gallery beginning Jan. 13. (photo courtesy of Vicente Siso)... |
| | Speech | 16.1 | 80 | Cheers, y'all. Now check it out. I really didn't even eat enough to be wiping my mouth, but I can tell you this, my mouth is salivating though. .... |
| | Literary | 19.0 | 80 | The advancement of Humanity never ceased, even for a moment–during difficult times we grow and adapt once again. The cities are as prosperous as ever, and our technological advancement is rising. ... |
| | Social | 39.9 | 22 | In general I really like the interplay between the two games. The advantage of shortform stories is that you can "skip to the good part"... |
| WMT 2025 Kocmi et al., 2025a | Dialogue | 9.08 | 191 | X: I am looking for a cheap hotel with free parking near Cambridge. Y: I have multiple cheap hotels with free parking. What part of town are you interested in staying in?... |
| | Literary | 14.4 | 121 | It had been a remarkable twenty-year pro career, one that most players could only dream of. He wore a gleaming championship ring, a testament to his hard work and dedication .... |
| | Social | 18.9 | 76 | Another fine evening (ok not really, it's wet and drizzly, but) to continue exploring my stash of Rum from Japan Cor Cor again - this time the "Industrial" .... |
| | News | 23.9 | 94 | Some folks really do deserve a badge of honour for their pedantry (C8). Veronica Coyne of Springfield claims that "when bemoaning the loss of the express lane at Woolies "12 items or less,"... |
| | Speech | 25.6 | 142 | Gotta watch a netflix show you feel me, but let me know down below. What show should i watch on netflix though? Because i'm i'm really having some trouble to find what show should ... |
| FLORES-101 NLLB Team et al., 2022 | Wikinews/politics | 3.9 | 22 | Mr Costello said that when nuclear power generation becomes economically viable, Australia should pursue its use.... |
| | Wikinews/sports | 4.2 | 19 | Mr Reid managed to drive the New Zealand's A1GP car, Black Beauty at speeds over 160km/h seven times over the bridge.... |
| | Wikinews/disasters and accidents | 4.9 | 18 | At 1:15 a.m. Saturday, according to witnesses, the bus was going through a green light when the car made a turn in front of it.... |
| | Wikivoyage/travel | 7.4 | 21 | Cold weather is perhaps the only real danger the unprepared will face.... |

| Our topics | Difficulty↑ | Words↓ | Example |
|---|---|---|---|
| Incarceration Prison vs Jail | 39.5 | 29 | Jails are short-term facilities for temporary detention, ... Prisons are long-term facilities for extended incarceration.... |
| Leasehold Estates Tenancy for Years Periodic Tenancy | 29.6 | 32 | Periodic Tenancy: A non-freehold estate that lasts only from period to period without having any definite duration that is longer than one .... |
| Future Interests Reversions Remainders Executory Interests | 29.5 | 34 | Future interests are legal rights to property ownership that may become possessory later. They arise when a grantor conveys.... |
| Removal Jurisdiction State to Federal Court | 21.2 | 30 | For removal based on diversity jurisdiction, the amount in controversy must exceed $75,000, and there must be complete diversity of .... |
| Victim Impact Statements Role | 21.0 | 34 | Victim impact statements detail the emotional, physical, and financial consequences of a crime. They can be written or oral and are... |

Table 5: From each domain in an existing benchmark, we selected the 25 most challenging sentences for three models: Google Translate, Gemma 3-27B, and Gemini 2.5 Pro. Results show that our algorithm's most challenging topic, "Incarceration: Prison vs. Jail," is comparable in difficulty to the most challenging subsets found across 12 domains in three widely used benchmarks. Generally, all five of the most challenging topics discovered by our algorithm are more difficult than most subsets of the three benchmarks.

| | |
|---|---|
| English→Czech (Google Translate) | **Topic:** Incarceration Prison vs Jail
**Text:** The key difference lies in the length of stay and jurisdiction. Jails are for temporary detention and operated locally, while prisons are for extended incarceration and managed by state or federal agencies.
**Translation:** Klíčový rozdíl spočívá v délce pobytu a jurisdikci. Věznice slouží k dočasnému zadržení a jsou provozovány na místní úrovni, zatímco věznice jsou určeny pro dlouhodobé věznění a spravují je státní nebo federální agentury.
**Quality estimation:** 40/100
**Error analysis:** The terms *jails* and *prisons* are both translated as *věznice* which is a critical error when the point of the text is to explain the differences. The correct translation for *jail* would be *vazba*. |
| English→Czech (Gemini 2.5 pro) | **Topic:** Preemption Doctrine Express vs Implied
**Text:** Congress sometimes expressly provides that state laws on a given topic are preempted (this is known as "express preemption"). Other times state laws are held preempted by courts because it is impossible to comply with both a federal and state law or because the state law frustrates accomplishment of the federal law's objectives (this is known as "implied preemption").
**Translation:** Kongres někdy výslovně stanoví, že státní zákony týkající se daného tématu jsou nadřazené (toto je známé jako „výslovná nadřazenost"). Jindy jsou státní zákony považovány za nadřazené soudy, protože není možné dodržovat jak federální, tak státní zákon, nebo protože státní zákon maří dosažení cílů federálního zákona (toto je známé jako „implicitní nadřazenost").
**Quality estimation:** 15/100
**Error analysis:** The term *preempted* (meaning *overriden*) is translated as *nadřazené* (meaning *superior*). This flips the meaning from the federal laws having priority to state laws having priority. The correct term is *přednost*. |
| English→Czech (Gemma 3) | **Topic:** Hearsay Evidence Admissibility Weight
**Text:** Hearsay is an out-of-court statement offered in court to prove the truth of the matter asserted. It is generally inadmissible unless it falls under an exception. The weight of hearsay evidence is determined by the trier of fact.
**Translation:** Doslovný výrok je mimožaludní prohlášení předložené u soudu k prokázání pravdivosti tvrzeného faktu. Obecně je nepřijatelný, pokud nespadá pod výjimku. Váha důkazů založených na doslovném výroku je určena soudem nebo porotou.
**Quality estimation:** 25/100
**Error analysis:** The term *hearsay* (meaning that the evidence can not be established) is translated as *doslovný* (meaning *literal*), which flips the meaning. The word *out-of-court* is translated as *mimožáludní* which is not a word in Czech. |

Table 6: Example texts (inputs) and translations from English→Czech based on found difficult topics. Only major errors are discussed.

**English→German (Gemini 2.5 pro)**

**Topic:** Removal Jurisdiction State to Federal Court

**Text:** Removal is the procedure that allows a defendant sued in state court to transfer the case to federal court if the plaintiff could have originally filed the case in federal court.

**Translation:** Unter Abschiebung versteht man das Verfahren, das es einem vor einem Staatsgericht verklagten Beklagten ermöglicht, den Fall an ein Bundesgericht zu überweisen, wenn der Kläger den Fall ursprünglich vor einem Bundesgericht hätte einreichen können.

**Quality estimation:** 25/100

**Error analysis:** The term *removal* is translated as *Abschiebung* (meaning *deportation*). The correct term is *Überweisung*.

**English→German (Google Translate)**

**Topic:** Incarceration Prison vs Jail

**Text:** Jails are typically locally operated, short-term facilities for those awaiting trial/sentencing or serving misdemeanor sentences (under a year). Prisons are state or federally operated long-term facilities.

**Translation:** Gefängnisse sind in der Regel lokal betriebene, kurzfristige Einrichtungen für Personen, die auf ihren Prozess/ihre Verurteilung warten oder eine Freiheitsstrafe für ein Vergehen (unter einem Jahr) verbüßen. Haftanstalten sind staatlich oder bundesstaatlich betriebene Langzeiteinrichtungen.

**Quality estimation:** 40/100

**Error analysis:** The terms *jails* and *prisons* are translated as *Gefängnisse* and *Haftanstalten*. While both of those terms could be jail and prison depending on the context, *Gefängnisse* implies a longer-term facility while *Haftanstalten* is used for shorter-term purposes. The intended meaning is ths reversed.

**English→German (Gemma 3)**

**Topic:** Hearsay Evidence Admissibility Weight

**Text:** Hearsay is defined as an out of court statement, made in court, to prove the truth of the matter asserted. In other words, hearsay is evidence of a statement.

**Translation:** Klatsch ist definiert als eine außergerichtliche Aussage, die vor Gericht gemacht wird, um die Richtigkeit des behaupteten Sachverhalts zu beweisen. Mit anderen Worten, Klatsch ist der Beweis einer Aussage...

**Quality estimation:** 25/100

**Error analysis:** The term *hearsay* (meaning that the evidence can not be established) is translated as *Klatsch* (meaning *gossip*). While informally this would be acceptable, it is improper terminology in this judicial context. The correct term is *Hörensagen*.

Table 7: Example texts (inputs) and translations from English→German based on found difficult topics. Only major errors are discussed.

| | |
|---|---|
| English→Chinese (Gemini 2.5 pro) | **Topic:** Preemption Doctrine Express vs Implied
**Text:** Express preemption occurs when Congress includes language in a federal statute explicitly stating that state law is preempted. An example is the Federal Aviation Administration Authorization Act of 1994.
**Translation:** 当国会在联邦法规中明确规定州法律被优先适用时，就会发生明示优先适用。例如1994 年的《联邦航空管理局授权法》。
**Quality estimation:** 45/100
**Error analysis:** The term *Express preemption* is translated as 优先适用 (meaning *applied with priority*). The correct term is 明示取代. |
| English→Chinese (Google Translate) | **Topic:** Incarceration Prison vs Jail
**Text:** Jails are local facilities confining persons before or after adjudication. Prisons are state/federal facilities confining convicted felons. Jails have diverse populations; prisons hold sentenced felons.
**Translation:** 监狱是地方性机构，用于关押判决前或判决后的人员。监狱是州/联邦机构，用于关押已定罪的重罪犯。监狱的关押对象多种多样；监狱关押的是已判刑的重罪犯。
**Quality estimation:** 40/100
**Error analysis:** The terms *jails* and *prisons* are both translated as 监狱. While 监狱 can sometimes encompass both concepts in a very general sense or be used colloquially, in a legal/technical context like this, distinguishing them is crucial. "*Jails*" are typically translated as 看守所. "*Prisons*" are correctly translated as 监狱 |
| English→Chinese (Gemma 3) | **Topic:** Battery Harmful or Offensive Contact
**Text:** Battery is the intentional act of causing harmful or offensive contact. Unlike assault (which can be just a threat), battery requires actual physical contact. This contact need not cause injury, only be offensive.
**Translation:** 电池是指故意造成有害或冒犯性的接触行为。与袭击（仅可以是威胁）不同，电池需要实际的身体接触。这种接触不一定需要造成伤害，只需具有冒犯性即可。
**Quality estimation:** 15/100
**Error analysis:** Mistranslation of "*Battery*" (legal term) as 电池 (electric battery): Critical/Major. At legal term, Battery means the completed act of unwanted physical contact. |

Table 8: Example texts (inputs) and translations from English→Chinese based on found difficult topics. Only major errors are discussed.

