# OpenReview forum: "Searching for Difficult-to-Translate Test Examples at Scale"
_ICLR.cc/2026/Conference — Submitted to ICLR 2026_

### Official Review · Reviewer_oSSB · 2025-10-31

**Soundness:** 3
**Presentation:** 3
**Contribution:** 3
**Rating:** 4
**Confidence:** 3

**Summary:**

Interesting research for searching difficult translation examples. In Natural Language Processing (NLP), finding difficult - to - translate test examples is crucial for effectively evaluating model capabilities. Since the Internet offers a vast amount of language data but manual curation is infeasible due to scale and topic diversity, this paper formalizes the task as a multi - armed bandit problem to identify the most challenging topics within a fixed computational budget for machine translation. Here, we will explore the advantages and disadvantages of this approach to understand its implications for NLP research.

**Strengths:**

Research strengths include:

- Efficient in Resource Allocation​

By formalizing the task of finding difficult - to - translate examples as a multi - armed bandit problem, the approach can efficiently identify the most difficult topics within a fixed computational budget. Each topic is treated as an “arm”, and pulling an arm involves drawing and evaluating an example. This way, it strategically explores the topic space, avoiding blind brute - force evaluation across all topics. For example, instead of exhaustively sampling from every potential topic, it focuses on the most promising ones, thus saving computational resources and time.​

- Scalable and Adaptable​

The method can generate and sample from a large amount of Internet data. It hierarchically generates topics starting from top - level ones and samples text through language model generation with the help of Google search tool calling. This process is generally inexpensive. Moreover, this formulation is compatible with any NLP task where Internet data can be a valid input and the input difficulty can be estimated. It can continuously explore new topics, adapting to different evaluation requirements in various NLP tasks, such as machine translation, question - answering systems, etc.​

- Outperforms Baseline Methods​

Experimental results show that various bandit strategies, like the ε - greedy algorithm, vastly outperform baseline methods such as brute - force searching. As shown in Figure 2, with the same budget and the cost of a single sampling being 1, the bandit - based search algorithms can achieve higher top - 1 and top - 10 difficulty scores. This indicates that these strategies can more effectively discover challenging test data for machine translation, providing better evaluation data for NLP models.

**Weaknesses:**

- High Computational Cost​
Each difficulty estimation of a source text involves generating a translation using a target model and then evaluating the translation’s quality to determine the text’s difficulty. For example, in the case of machine translation, this process requires significant computational resources. Since the number of potential topics is vast and each topic may need to be sampled multiple times, the overall computational cost can be extremely high. This high cost limits the scalability of the approach, making it challenging to apply on a large - scale, especially when dealing with a large number of topics and models.​

- Difficulty in Topic Selection​
Judging the actual difficulty of a topic for a model is challenging. There is a disconnect between a topic’s perceived difficulty by a human and its actual difficulty for a model. For instance, a topic that seems complex to humans, like Baroque music theory, might be easily processed by a model, while a seemingly simple topic like concrete masonry can be difficult for a model if its specialized terminology was not present in the training data. This makes it hard to accurately select the most difficult topics, and incorrect topic selection can lead to wasted resources and less - effective evaluation data.​

- Reliance on Model and Estimation Method​
The approach relies on a specific set of models for translation and a particular method for difficulty estimation. Different models may produce different translation results, and different difficulty estimation methods may yield varying difficulty scores for the same text. This means that the results obtained are highly dependent on the chosen model and estimation method. As a result, the generality and reliability of the discovered difficult - to - translate examples may be affected. If the model or the estimation method changes, the identified difficult topics and examples may also change, reducing the stability and broad applicability of the findings.

**Questions:**

Providing computational cost​ comparements and reducing methods , this research score will be much higher.
Topic selection is interesting, but only providing cases. More analysis will be helpful.

---

> ### Author Response · Authors · 2025-11-23
>
> Thank you for your review! We wish to address the listed weaknesses:
>
> > High Computational Cost​
>
> Mitigating the computation costs at locating the challenge source texts **is exactly the motivation and goal of our paper**. Given the vast search space and the long-tail distribution of challenge examples, a brute-force approach is computationally intractable. Our method addresses this by drastically increasing efficiency. Specifically, we showed (Line 213) that we only require computation on 6% of the entire search space (corresponding to "less than 2 samples per topic," Line 212). This massive decrease in needed difficulty estimations results in an overall cost reduction of **100-fold** (see Table 3).
>
> > Difficulty in Topic Selection​
>
> In our setup, we study the difficult for models, not for humans (Line 83). Our objective is to find failure modes in high-performing models and curate a test set that provides a strong hill-climbing signal for development. Therefore, the potential misalignment between human and model difficulty estimation is not a concern for this paper.
>
> > Reliance on Model and Estimation Method​
>
> We agree that this could be a big potential concern but **for this reason in Section 3.6 we show evidence that examples that were found to be difficult for one model are difficult for other models**. Specifically, the Top 10 challenge topics for Gemini, Gemma, and Google Translate models are all found within the Top 50 challenge topics (out of 3200) for each of the other models.
>
> Please let us know if this sufficiently addresses your concerns. We are happy to elaborate more!

---

### Official Review · Reviewer_Qabr · 2025-10-31

**Soundness:** 3
**Presentation:** 3
**Contribution:** 2
**Rating:** 4
**Confidence:** 3

**Summary:**

This paper introduces a framework for automatically discovering challenging test examples for machine translation by framing the search process as a multi-armed bandit problem. The goal is to efficiently identify the most difficult topics within a fixed computational budget. The paper demonstrates that bandit-based strategies (especially ε-greedy) significantly outperform brute-force baselines, scaling to millions of topics while uncovering test sets that are more challenging than existing benchmarks like WMT and FLORES. The approach is task-agnostic and could extend to other NLP tasks.

**Strengths:**

1. I like the experiment  on the scalability an efficiency, which empirically validates that ε-greedy achieves near-oracle performance with only ~6% of the budget required for brute-force evaluation, and  Demonstrates logarithmic scaling of difficulty with topic set size, supporting theoretical claims. Also the error analysis reveals terminology/accuracy errors (70% major errors), aligning with known MT weaknesses.

**Weaknesses:**

1. I feel the comparison with other baselines is missing, such as active learning or curriculum learning baselines, which also target difficult examples.
2. I would like to see the generalization of this frame across different tasks. While framed as task-agnostic, experiments are exclusive to MT. No validation on other NLP tasks (e.g., summarization, QA) is provided, weakening claims of broader applicability.

**Questions:**

see weakness for detail:  comparison with active learning or curriculum learning baselines and the generalization across different tasks.

---

> ### Author Response · Authors · 2025-11-23
>
> Thank you for your review! We hope to address your concerns:
>
> 1. To our understanding, curriculum learning is an applications that make use of our contributions (i.e. sort by increasing difficulty etc) and active learning is already happening with our approach as it is interactive and we update our distribution based on each sample. We would be delighted if you could elaborate on what baselines specifically you have in mind.
> 2. We are adding an English based factual question answering task in both closed and open book settings (the specific setup is similar to [ECLeKTic benchmark](https://arxiv.org/pdf/2502.21228)). We hope to add the results in the coming days.
>
> Thank you for your time. We are happy to elaborate further and engage in discussion!

---

> ### Author Response · Authors · 2025-11-24
> **Adding factual question answering task**
>
> Per your suggestion in number 2, we applied our setup to a knowledge question answering task (L702-L748).
>
> We find the following:
> - Knowledge QA trends align closely with the translation task results. Bandit-based algorithms greatly outperform the brute-force approach, further demonstrating the generalizability of our framework.
> - The $\epsilon$-greedy demonstrates superior performance over other bandit approaches.
> - Most importantly, we recover the oracle difficult samples by exploring only 10% of the total search space (13k out of 130k subtopics), equating to fewer than 3 samples per domain.

---

### Official Review · Reviewer_r5Fe · 2025-11-01

**Soundness:** 3
**Presentation:** 3
**Contribution:** 3
**Rating:** 8
**Confidence:** 4

**Summary:**

This paper addresses the challenge of efficiently identifying difficult test data for language models. The authors formalize the search for challenging topics as a *multi-armed bandit* problem, where each topic represents an "arm," and sampling it (at a cost) produces an example whose difficulty is then evaluated. The objective is to identify the most difficult topics within a limited computational budget. Applied to machine translation, the proposed method significantly outperforms brute-force or greedy approaches in discovering challenging test examples.

**Strengths:**

The paper addresses an important practical problem: how to effectively identify the weaknesses of language models. The problem formulation and the final solution are neat and well-suited to the task. The efficacy of the proposed method is further supported by comprehensive empirical studies.

**Weaknesses:**

1. It would be more interesting if the paper can include one more task, such as safety or factuality. I believe the post-training stage shares similar problems, e.g., designing a prompt set for the reinforcement fine tuning step.
2. Minor comments:
    - "desiderate" &rarr; "desiderata" in line 54.
    - Avoid using the same notation to represent different concepts. For example, $t$ is used for both "translation" (line 84) and "topic (throughout the rest of the paper).
    - Hyphens in "top-k"s are incorrectly in math mode.

**Questions:**

See ``Weaknesses``.

---

> ### Author Response · Authors · 2025-11-23
>
> Thank you for your review and you ! We hope to address your concerns:
>
> 1. We are adding an English based factual question answering task in both closed and open book settings (the specific setup is similar to [ECLeKTic benchmark](https://arxiv.org/pdf/2502.21228)). We hope to add the results in the coming days.
> 2. Thank you for pointing out the typesetting issues, which we have fixed since then.
>
> Thank you for your time. We are happy to elaborate further and engage in discussion!

---

> ### Author Response · Authors · 2025-11-24
> **Adding factual question answering task**
>
> Per your suggestion, we applied our setup to a new task (knowledge question answering) in L702-L748.
>
> We find the following:
> - Knowledge QA trends align closely with the translation task results. Bandit-based algorithms greatly outperform the brute-force approach, further demonstrating the generalizability of our framework.
> - The $\epsilon$-greedy demonstrates superior performance over other bandit approaches.
> - Most importantly, we recover the oracle difficult samples by exploring only 10% of the total search space (13k out of 130k subtopics), equating to fewer than 3 samples per domain.

---

### Official Review · Reviewer_Xqq7 · 2025-11-01

**Soundness:** 2
**Presentation:** 2
**Contribution:** 1
**Rating:** 2
**Confidence:** 4

**Summary:**

This paper introduces a MAB framework for automatically identifying difficult topics from large-scale Internet text to construct more challenging test sets for NLP models, particularly machine translation. The authors observe that example difficulty is stochastically related to its topic, some topics tend to yield harder examples, but this relationship is noisy and costly to explore exhaustively across thousands of potential topics. To address this, the paper formalizes topic selection as a stochastic bandit problem, where each topic is treated as an arm, and pulling an arm corresponds to sampling a text from that topic, translating it, and measuring its difficulty using a model-based quality estimation metric (e.g., GEMBA). Within a fixed query budget, the goal is to identify the k most difficult topics efficiently.
The paper proposes and compares several bandit strategies (random, greedy, and epsilon-greedy, with batched variants) for this exploration-exploitation trade-off, showing that bandit methods significantly outperform uninformed or brute-force baselines in finding difficult topics.

**Strengths:**

* The paper's framing of difficult-topic discovery as a multi-armed bandit problem is intuitive and useful.

**Weaknesses:**

* The conceptual novelty is limited. The formulation is a straightforward application of standard stochastic MAB principles, with no new theoretical insights, regret analysis, or algorithmic extensions.
* The difficulty estimation pipeline and sampling procedures rely entirely on existing methods, and the empirical results confirm expected behavior rather than reveal new dynamics.
* Furthermore, the approach ignores the inherent structure of the topic space, missing opportunities for contextual or hierarchical modeling. Overall, the contribution is incremental, a solid engineering integration rather than a substantive methodological or theoretical advance not suitable for ICLR.

**Questions:**

Kindly refer to weaknesses

---

> ### Author Response · Authors · 2025-11-23
>
> Thank you for your review! We hope to address your concerns:
>
> > The conceptual novelty is limited. \
> > rely entirely on existing methods
>
> We believe there is a misunderstanding regarding the scope of our contribution. Our core contribution is *not* an algorithmic invention, but rather the **formulation of a critical NLP problem (finding failure modes in high-performing models and curating a testset that reflects the strong hill climbing signals)**. Standard benchmarks (e.g., WMT, FLORES) are saturated, yet improving these models requires identifying the remaining tail of failure modes, which static benchmarks don’t capture. In this work we shift the evaluation paradigm from static curation to dynamic, which provides a "hill-climbing" signal for model development. We are the first work to formulate finding the difficult to translate topics on the Internet as a multi armed bandit problem. The fact that standard algorithms (like $\epsilon$-greedy) function efficiently in this domain is also a novel finding which allows us to solve a real problem at the fraction of the cost of current go-to approach.
>
> > ignores the inherent structure of the topic space, missing opportunities for contextual or hierarchical modeling
>
> We address this in Appendix B.1 and Algorithm 5 , where **we explicitly implemented a Contextual Multi-Armed Bandit to use the topic space structure**. However, we found that semantic similarity is not indicative of similarity in topic difficulty. Because the contextual approach added complexity without meaningfully outperforming the $\epsilon$-greedy baseline, we excluded it from the main results.
>
> Thank you for your time. We are happy to elaborate further and engage in discussion!

---

### Author Response · Authors · 2025-11-28
**Summarization of our rebuttal discussion**

We sincerely appreciate the reviewers’ comments and the time spent evaluating our work. We are encouraged that Reviewers Xqq7 and r5Fe recognized our problem formulation as both neat and practical. Qabr further confirms that our empirical results clearly support theoretical claims and error analysis reveals terminology/accuracy errors (70% major errors), aligning with known MT weaknesses from prior works. All four reviews highlighted the strength of our empirical results and recognized the high efficiency and practical implementation of our pipeline.

Regarding the specific points raised, we clarified to Reviewer Xqq7 that our primary contribution is the novel formulation of a critical NLP problem: dynamically finding failure modes in high-performing models to curate test sets that reflect strong hill-climbing signals. We are the first work to formulate finding the difficult to translate topics on the Internet as a multi armed bandit problem. The fact that standard algorithms (like epsilon-greedy) function efficiently in this domain is also a novel finding and immensely pracitally useful. We are pleased that these contributions were recognized by the other three reviewers. We provided Reviewer oSSB with further clarification regarding computation costs, the definition of topic difficulty, and the generalizability of our pipeline.

Most importantly, to address the generalizability concerns from Reviewers r5Fe and Qabr, we spent the rebuttal period **extending our framework to another task, Factual Question Answering**. The trends in QA align closely with the translation task results, with bandit-based algorithms massively outperforming brute-force approaches. Most importantly, we were able to recover the "oracle" difficult samples by exploring only 10% of the total search space (13k out of 130k subtopics), which equates to fewer than three samples per topic. We have also clarified the relationship between our pipeline and both curriculum learning and active learning to demonstrate that our paper contains sufficient baselines to support our claims.

We strongly believe these revisions and additional experiments have addressed all the reviewers' concerns.

---

### Meta-Review · Area_Chair_cVTC · 2025-12-19

**Summary:**

This work addresses the challenge of efficiently identifying difficult test data for language models by introducing a framework to automatically discover difficult examples. The discovery is formulated as a multi-armed bandit problem in searching, where each topic represents an arm and sampling it (at a cost) produces an example whose difficulty is then evaluated. The goal is to efficiently identify the most difficult topics within a fixed computational budget. The author showed efficacy of proposed approaches by comparing the bandit-based strategies with brute-force baselines, and observed performance gain on machine translation tasks including benchmarks WMT and FLORES. Arguably the approach is task-agnostic and could extend to other NLP tasks.

Reviewers found the paper is of following strength
* The work proposed an efficient way to discover difficult data examples with a reasonable problem formulation
* The paper select reasonable benchmarks to demonstrate the effectiveness of proposed method.

**Reviewer Concerns:**

We also found the additional explanation from authors during rebuttal are helpful (e.g., discussing reliance on models, adding more evaluation tasks, explaining more methodology details and insights). These are very helpful experiments/discussion and will strengthen this work.

Although this submission is of reasonable quality and contribution to science community, the reviewers found impact of this work is at the borderline compared to a typical ICLR paper given the work focuses on a specialized subarea (data selection for improving machine translation, or a bit extension in natural language understanding). Given that nowadays LLMs leverages aggressive scaling (compute, training data) during pretraining to boost text understanding performance across all tasks, impact seems slightly narrow to focus on improving certain task on top of foundation models. This work seems more relevant to NLP focused conferences and thus we encourage the authors to resubmit.

**Reviewer Scores:**

see above

---

### Decision · Program_Chairs · 2026-01-26

Reject